# Design, Synthesis, and Evaluation of Near-Infrared Fluorescent Molecules Based on 4H-1-Benzopyran Core

**DOI:** 10.3390/molecules26226986

**Published:** 2021-11-19

**Authors:** Shuting Wang, Shulei Zhu, Yawen Tanzeng, Yuexing Zhang, Chuchu Li, Mingliang Ma, Wei Lu

**Affiliations:** 1Shanghai Engineering Research Center of Molecular Therapeutics and New Drug Development, School of Chemistry and Molecular Engineering, East China Normal University, Shanghai 200062, China; wangshu_ting_1@163.com (S.W.); slzhu@chem.ecnu.edu.cn (S.Z.); 51204300157@stu.ecnu.edu.cn (C.L.); wlu@chem.ecnu.edu.cn (W.L.); 2School of Ecological and Environmental Sciences, East China Normal University, Shanghai 200241, China; 3Key Laboratory of Brain Functional Genomics-Ministry of Education, School of Life Science, East China Normal University, Shanghai 200062, China; 10194304411@stu.ecnu.edu.cn; 4Collaborative Innovation Center for Advanced Organic Chemical Materials Co-Constructed by the Province and Ministry, Ministry of Education Key Laboratory for the Synthesis and Application of Organic Functional Molecules, College of Chemistry and Chemical Engineering, Hubei University, Wuhan 430062, China; zhangyuexing@sdu.edu.cn

**Keywords:** dye, fluorescence intensity, environmental sensitivity, near-infrared fluorescent molecules, potential probe

## Abstract

A series of novel fluorescent 4H-1-benzopyrans was designed and developed as near-infrared fluorescent molecules with a compact donor–acceptor-donor architecture. Spectral intensity of the fluorescent molecules **M-1, M-2, M-3** varied significantly with the increasing polarities of solvents, where **M-3** showed high viscosity sensitivity in glycerol-ethanol system with a 3-fold increase in emission intensity. Increasing concentrations of compound **M-3** to 5% BSA in PBS elicited a 4-fold increase in fluorescence intensity, exhibiting a superior environmental sensitivity. Furthermore, the in vitro cellular uptake behavior and CLSM assay of cancer cell lines demonstrated that **M-3** could easily enter the cell nucleus and bind to proteins with low toxicity. Therefore, the synthesized near-infrared fluorescent molecules could provide a new direction for the development of optical imaging probes and potential further drugs.

## 1. Introduction

Since British physicist Joseph John Thomson discovered free electrons from cathode rays, the application of electrons has brought earth-shaking changes to human life [1]. Especially in the booming period of chemistry and industry, electrons have played an indelible role in advancing the development of chemical materials [2,3,4]. Precisely due to the unique characteristics of electrons, building the construction of D-A-D conjugated light-emitting structures has propelled the advancement of fluorescent small molecules. When it comes to D-A-D or D-A structure, electrons transition freely in the entire conjugation interval, and as the effective conjugation area expands, the energy band gap decreases, which would change the fluorescence characteristics of target compounds. Therefore, through the modification of the conjugated skeleton or terminal substituents, different application requirements of fluorescent molecules could be met [5]. Currently, D-A or D-A-D fluorescent molecules including rhodamine, fluorescein and BODIPY [6,7,8,9,10,11] have several main applications, such as detecting diseases (cancers and Alzheimer) [12,13], imaging live-cell fluorescence [14], and being fluorescent molecular switches [15]. In this work, we laid focus on the coumarins [16], and explored the changes in the fluorescence properties of a series of compounds when the electron donor and acceptor structure are changed. When it comes to luminescence mechanism, there are mechanical luminescence (including that triboluminescence and mechanical force is sufficient to break the material bonds and cause fractures, which called fractoluminescence [17]) or photoluminescence [18].

Although the original coumarin fluorescent molecule usually presents a low two-photon absorption cross section [19], the absorption can be increased with the expansion of the conjugate system. Meanwhile, the introduction of electron-donating groups (EDG) or electron-withdrawing groups (EWG) would bring about large two-photon absorption (TPA) in the near-infrared reflectance (NIR) region of fluorescent molecules (Figure 1). This structure will show potential applications in biology, especially in photoacoustic imaging or ion detecting [20].

In this work, dimethyl group was introduced at 1-position of the coumarin core, and p-π hyperconjugation expanded with the increase of electron donating part. In addition, the introduction of vinyl chain [21] at position 3 could effectively expand the conjugation area, and the para-substituted benzene ring can also bring electron-withdrawing groups. Under the guidance of the above ideas, a series of fluorescent molecules **M-1, M-2, M-3** were rationally designed, synthesized (Figure 1) and modified with molecular weight less than 400, realizing the leap from far infrared to infrared region [22]. It is worth mentioning that this series of compounds exhibited excellent solvent sensitivity [23,24] and superior quantum yield [25,26] in a hydrophobic environment. When applied to live-cell imaging, these fluorescent molecules could easily reach the nucleus, suggesting the huge potential applications for early diagnoses.

## 2. Results

### 2.1. Chemical Properties Experiment

#### 2.1.1. Spectral Properties of Compound **M** Series

To investigate the spectroscopic properties of compounds **M**, we calculated their π−π* transitions using time-dependent density functional theory [27]. According to the geometrical parameters of the optimized ground state of the 3 compounds, the S1 state of all compounds belongs to the charge transfer singlet state 1CT, which is mainly formed by the transition from HOMO to LUMO. For the D-π-A system, the electron cloud of HOMO was mainly delocalized on donor part and the molecular core, while LUMO is completely delocalized on the acceptor plane. The HOMO and LUMO electron clouds of checked the three compounds are completely separated, and the numerical difference is tiny; also, the degree of separation of the electron clouds between HOMO and LUMO is greater than that of existing fluorescent small molecules. The overlap of electron cloud between HOMO and LUMO is mainly determined by the steric hindrance of the atoms connected between the donor part, molecular core and acceptor unit [27]. The greater the steric hindrance is, the greater distortion degree is between the units, and the same as conjugated π bond. The smaller the angle became, the greater the degree of separation of the front-line orbital electron cloud is [28,29]. It can be seen from Figure 2A,B that the D-π-A **M** series compounds basing on the coumarin core with a specific steric hindrance (four-membered, five-membered, six-membered heterocyclic ring containing N), where the electron clouds of these compounds’ HOMO and LUMO (Appendix A) can be effectively separated. This structural feature is conducive to small ΔE_st_ [30]. Therefore, these three compounds have great potential to create more near-infrared fluorescent molecules probe.

The spectral properties of the **M** series were observed in various solvents. According to UV absorption spectrum data (Table 1), the targeted compounds have tiny different max absorption wavelengths. The main absorption peak of the synthesized compounds ranks from 480 nm to 520 nm. With the distinguishment in donor part of the whole p-conjugated bridge of D-A-D architecture, there is a wider Stokes Shift with the increasing rigidity of nitrogen-containing cycloalkane. It was rationally predicted that the rigidness has an impact on the rotation of carbon-carbon single bond and fixed conjugate delocalization range, leading to strong fluorescence characteristics. Therefore, **M-3** with the most excellent fluorescence characteristics is selected from the three compounds as the model molecule for the following discussion.

The quantum yield test in this experiment obtains Q_Ein_ (Internal Quantum Efficiency) through an integrating sphere device, The AF value is a calibration parameter of the instrument and does not affect the comparison of internal quantum yield.

The optical properties of compound **M-3** were observed in several solvents with diverse polarities. The detailed wavelength of absorption, excitation, and emission maxima and the Stokes shift of **M-3** in different solvents were shown in Table 1. **M-3** showed the highest optical intensity in DMSO. Moreover, with the increasing polarity of solvents, the optical intensity of **M-3** enhanced [31]. Expectedly, **M-3** showed no optical ability in PBS and H_2_O, suggesting a greater stabilizing effect on the excited state than on the ground state [32].

The maxima absorption wavelength in PBS is at 486 nm, while the maxima in the other solvents were found in the range of wavelengths 490–508 nm (Figure 3A and Table 1). Stokes shift of **M-3** are in the range of wavelengths 275–320 nm, which is a typical Donor-Acceptor-Donor (D-A-D) fluorescent molecule. If the solvents are more polar than acetone, the existence of intramolecular charge transfer state got through nonradiative deactivation, causing the quench of fluorescent emission of **M-3** (Figure 3B). **M****-3** didn’t fully overcome aggregation quenching in water and PBS, since fluorescence emission intensity was much weaker than the emission intensity in organic solvents. Though quenching wasn’t completely overcame, but that did not affect the entry of **M-3** into the cell to bind to proteins in further tests.

#### 2.1.2. Viscosity Sensitivity Studies of **M-3**

Figure 4 showed the emission behavior of **M-3** at eleven different viscosities. In the experiment, PEG-400 was used as a viscous solvent, and the concentration of PEG-400 ranges from 0% to 100%, divided into gradient concentrations. PEG-400 was mixed with ethanol in proportion to determine the corresponding emission spectrum of **M-3**. The concentration of **M-3** maintained below 50 nM in order to reduce the possibility of hydrogen bonding, aggregation and self-quenching [33]. As the concentration of PEG-400 increased, the increase in fluorescence can be clearly observed (Figure 4A). The spectrum results in Figure 4 showed that the fluorescence intensity of **M-3** had increased by approximately 2 times, implying its viscosity sensitivity. The UV spectroscopy and fluorescence emission spectra data of M-2 and M-3 in various solvents were showed in Appendix A for comparison and reference It was known that the polarity of the solvent affects the emission intensity of the emission intensity of the charge transfer class of molecules. Therefore, the effects of increased polarity of protons and polar solvents (such as EtOH and PEG-400) need should to be considered [34]. Also, the luminescence mechanism of **M-3** was known as a single bond rotation as well as delocalization of π electrons in the conjugate region. Herein, PEG-400 played a great role in increasing the viscosity of the medium to fix the rotation strength, so that electrons applied activities in a relatively regular space.

#### 2.1.3. BSA Binding Affinity of **M-3**

According to the previous viscosity sensitivity studies, data showed **M-3** can penetrate the cell membrane inside the cell, and the binding affinity of **M-3** in bovine serum albumin was further validated in order to show the efficient binding to the protein after entering the cell. BSA was dissolved in PBS to prepare 3% BSA solution, and **M-3** was dissolved in DMSO to prepare a 10 mM mother liquor. After calculating the equivalent, the **M-3** mother liquor was added to BSA solution to prepare a gradient concentration of BSA+**M-3** mixed solution. It is worth mentioning that after full mixing each time, the mixed solution needed to be left still at room temperature for 20–40 min, and finally the fluorescence absorption-emission spectrum measurements were performed. After mixing with BSA, **M-3** bound to the protein and the molecular structure was fixed. Since there was gaps between molecules, they no longer gathered together. Moreover, due to single bond rotation decreased and the electronic transition area tended to be fixed, there were stable fluorescence emission wavelength measured. With the increasing concentration of **M-3**, fluorescence intensity in BSA was caught a linear increase and a mild blue shift was observed in the emission wavelength (Figure 4B). The fluorescence emission intensity of the 100 nM **M-3** group was over 4-fold stronger than that of the control group, suggesting that **M-3** showed significant environmentally sensitivity in living cells, which led to a solid basis for further biological evaluations.

#### 2.1.4. Fluorescence Lifetime Measurements of **M-3**

Related fluorescent molecule properties are mainly governed by the excited state bond twisting or rotation, leading to non-radiative decay from the excited state back to the ground state.

First, as shown in Figure 5A, fluorescence decays of **M-3** in H_2_O, PBS, MeCN, MeOH, EtOH, EA, and DCM were measured. An increase in the fluorescence lifetime was observed from 0.2 ns to 0.9 ns (Appendix A) with the increase of solvent polarity. It could be explained by the stabilization of the TICT state due to strong solvation in a more polar medium, which leads to an increase in lifetime and internal quantum efficiency (Table 1 and Appendix A). The fluorescence decay data of M-1 and M-2 in different solvent were showed in Appendix A. In addition, due to the hydrogen bond in water and alcohol [35], it will excite multiple vibrational quanta to reduce the fluorescence lifetime, leading to the relatively shorter value in H_2_O and PBS. Moreover, higher solvation in polar solvents especially the existing of hydrogen bonding in water and alcohol results in the effective transformation of excitation energy into multiple vibrational quanta, which in contrast would decrease the values of the fluorescence time.

Viscous solvents are used to hinder the rotation of molecules, to block the TICT state and the lifetime of **M-3** was proved to increase from 0.6 ns at 0% PEG400 to 1.4 ns at 100% PEG400, indicating the viscosity sensitivity of **M-3** (Figure 5B and Appendix A).

We also studied the fluorescence decay of the compound **M-3** in the presence of 3% BSA (Figure 5C), comparing with the decay behavior of **M-1, M-2** (Appendix A). The existence of BSA increased the fluorescence lifetime five times compared with organic solvents. This obvious enhancement is caused by the affinity binding of the compound **M-3** to the hydrophobic pocket of the protein. Due to the addition of BSA, small molecule **M-3** is difficult to aggregate after binding proteins, resulting in emission efficiency and enhancement of fluorescence lifetime.

### 2.2. Brief Biological Evaluation

HSC cell, known as healthy human liver stellate cell, is used here as the target of cytotoxicity tests. (Figure 6) HSC cells were treated with different concentrations of **M-3**, ranking from 10 to 100 M. Results of CCK-8 assay illustrated that all showed high cell viability over 92%, which indicated that **M-3** has low toxicity in healthy cells, indicating a great potential to create future fluorescent probes.

Based on the results discussed above, we evaluated that the tumor bioimaging capability of **M-3** in DU-145, HT-29, and SJSA-1 cell lines through a single-photon laser confocal microscope. Cells with proper densities were incubated with 50 μM of **M-3** at 37 °C under 5% CO_2_ for 24 h. As shown in Figure 7, all groups were observed to have strong red fluorescence in the cytoplasm, suggesting the tumor selective bioimaging ability.

To further confirm the subcellular distribution of **M-3**, after uptake into tumor cells, DAPI was used, which is known to stain intact nuclei selectively and strongly. The subcellular distribution behavior of **M-3** in three cells was measured by CLSM. As shown in Figure 7, strong red and blue fluorescence was observed in the nuclei, obviously demonstrating that **M-3** was mostly located in the nuclei. In short, all the results demonstrated the effective cellular uptake of **M-3.**

## 3. Discussion

In summary, a series of small fluorescent molecules based on 4H-1-benzopyran core was rationally designed and synthesized. Among them, **M-3** shows the best fluorescent characteristics. **M-3** exhibited long wavelength, strong red-emission, and highly efficient optical performance, due to the p···π and CH···N hyperconjugation effect [36]. Besides, **M-3** exhibited typical properties of molecular rotors with high viscosity sensitivity in a glycerol-ethanol system and showed high environmental sensitivity [37,38] in different polar solvents with a certain degree of polarity dependence. **M-3** also possessed high environmental sensitivity in that the addition of gradient concentrations of BSA in PBS and elicited a significant 4-fold increase in fluorescence intensity. Fluorescence lifetime measurements further confirmed the viscosity and BSA sensitivity properties. In addition, **M-3** showed obvious cellular uptake behavior in the three tumor cell lines, and it could smoothly enter the nucleus. Also, experimental data confirmed the possibility of using **M-3** in tumor imaging. Above all, all these results elucidate that **M-3** could be used for imaging of the tumor micro-environment and the detection of cancer lesions, which could also be conjugated to different high-affinity ligands for investigating various in vivo process. Furthermore, the benzene ring of the mother nucleus in the structure of **M-3** could be used for structural modification; if it was connected to a specific target head, it was expected to become a potential near-infrared probe.

## 4. Materials and Methods

### 4.1. Experimental Material

Bovine serum albumin (BSA) and PEG400 were obtained from Shanghai Macklin Biochemical Co., Ltd. (Shanghai, China). The other reagents used were all spectroscopic grade. UV–visible (UV–Vis) absorption spectra and emission spectra and fluorescence spectra were determined at room temperature (22–25 °C) at concentrations around 50 μM with SHIMADZU UV-3600 Plus Spectrophotometer (Kyoto, Japan) and Edinburgh Instruments FLS980 fluorescence spectrometer Spectrophotometer (Shanghai, China) with slit widths routinely set at 5 nm respectively. The quantum yield test in this experiment obtains Q_Ein_ data through an integrating sphere device, specifically an absolute quantum efficiency test integrating sphere. The test sample cuvette is placed in the center of the integrating sphere, and a beam of excitation light is used to directly illuminate the sample. The stimulated light emitted by the sample is received by the integrating sphere and transmitted to the optical measuring device to obtain the absolute quantum efficiency test result. The instrument was SHIMADZU UV-3600 Plus Spectrophotometer and Edinburgh Instruments FLS980 fluorescence spectrometer and integrating sphere device. The AF value is a calibration parameter of the instrument and does not affect the comparison of internal quantum yield. Fluorescence lifetime was detected on A1 fluorescence lifetime microscope system (Nikon, Tokyo, Japan). High performance liquid chromatography (HPLC) analysis was performed at room temperature using Nexera UHPLC LC-30A (Shimadzu, Kyoto, Japan). All images were mounted and observed with a LEICA TCS SP8 Confocal Microscope System Spectrophotometer (Weztlar, Germany). NMR spectra were recorded on a Bruker DRX-400 MHz spectrometer Spectrophotometer (Zurich, Switzerland). Chemical shifts were reported in ppm and coupling constants (J) were reported in Hz. High-resolution mass spectra (HRMS) were performed on an electron spray injection (ESI) Thermo Fisher Scientific LTQ FTICR mass spectrometer.

All calculations were performed using Gaussian 16 software Spectrophotometer (Wallingford, CT, USA). B3LYP functional, 6-31G(d,p) basis set, and IEFPCM solvent model were used to optimize the molecular structure and calculating properties in different solvent. TD-B3LYP were used to calculate the excitation energy and optimize the structure of the first excited state. For **M-1**, **M-2** and **M-3**, other functionals such as APFD, PBE1PBE, M06-2X, and wB97XD were also used to study the influence of functional [39].

### 4.2. Synthetic Procedures for Compounds ***M-2*** Compounds

General procedures for the preparation of compound **2**. NaH (60%, 5 g, 125 mmol, 5.0 equiv.) was dissolved in 40 mL tetrahydrofuran at 0 °C. After the mixture being stirred for 10 min, 1-(2-hydroxyphenyl)ethanone, (3.07 mL, 25 mmol, 1.0 equiv.) and ethyl acetate (7.4 mL, 75 mmol, 2.5 equiv.) were generally added in the above mixture, and then was 15 mL tetrahydrofura. The whole mixture was heated under reflux (65 °C) for 8 h. The mixture was cooled down to room temperature and 4M HCl was added dropwise until pH = 6–7. After well stirred, the mixture was extracted with ethyl acetate (3 × 85 mL). The extracted organic layer was then dried over Na_2_SO_4_, filtrated, and concentrated to dryness. The crude product was purified by column chromatography on silica gel (petroleum ether: ethyl acetate = 10: 1 to 8: 1) to give compound **2** as a cream-colored solid (3.1 g, 72%).

General procedures for the preparation of compound **3**. Compound **2** (1.5 g, 8.42 mmol, 1.0 equiv.) was dissolved in acetic acid (18 mL). After the mixture being stirred for 10 min, 98% H_2_SO_4_ (0.2 mL, 0.1 eq.) was gently added to the above solution. The mixture was heated under 85 °C for 45 min and poured into 120 mL ice water. Saturated sodium hydroxide solution was then added dropwise into the water solution until pH = 6–7. After well stirred, the mixture was extracted with ethyl acetate (3 × 90 mL). The extracted organic layer was washed by saturated NaCl solution (65 mL), then dried over Na_2_SO_4_, filtrated, and concentrated to dryness. The crude product was purified by column chromatography on silica gel (petroleum ether: ethyl acetate = 5:1) to collect compound **3** as white solid (1.1 g, 83%).

General procedures for the preparation of compound **4**. Compound **3** (1 g, 6 mmol, 1.0 equiv.) was dissolved in 15 mL acetic anhydride and stirred for 15 min. Propanedinitrile (0.59 g, 9 mmol, 1.5 eq.) were generally added in the mentioned solution. The reaction mixture was heated under reflux (110 °C) for 16 h. Then 1.5 mL water was poured into the cooled mixture and the mixed solution was further heated for 40 min. The reaction mixture was cooled to room temperature and filtered. The solution was concentrated and the crude product was purified by column chromatography on silica gel (petroleum ether: dichloromethane = 5:1) to collect compound **4** as mild yellow solid (0.77 g, 62%).

General procedures for the preparation of compound **M-1**, **M-2**, and **M-3**. Compound **4** (0.2 g, 0.96 mmol, 1.0 equiv.) was dissolved in 10 mL methylbenzene, and then 0.5 mL of piperidine and 0.5 mL of acetic acid were successively added. The well-mixed solution was heated under reflux (90 °C) for 12 h. The reaction mixture was cooled down to room temperature and filtrated. **M-1** was collected as red solid (0.15 g, 42%) after purified by column chromatography on silica gel (petroleum ether: ethyl acetate = 3: 1 to 1: 1); **M-2** was collected as purple solid (0.14 g, 40%) after purified by column chromatography on silica gel (petroleum ether: ethyl acetate = 2:1); **M-3** was collected as dark red solid (0.12 g, 36%) after purified by column chromatography on silica gel (petroleum ether: ethyl acetate =2:1). NMR spectra for M series was showed in Appendix A; The HR-MS and HPLC spectrum for M series was showed in Appendix A.

### 4.3. Cell Culture

Human healthy liver stellate cells (HSC), human osteosarcoma cells SJSA-1 cells, human colorectal adenocarcinoma HT-29 cells and human prostate cancer DU-145 cells were purchased from the cell bank of the Chinese Academy of Sciences (Shanghai, China). HSC cells and DU-145 cells were cultured in RPMI-1640 media; SJSA-1 cells were cultured in DMEM media; HT-29 cells were cultured in McCoy’s 5A medium. All media were supplemented with 10% (*v*/*v*) fetal bovine serum (FBS) and 1% (*v*/*v*) penicillin/streptomycin. Cells were incubated in a humid atmosphere of 5% CO_2_ at 37 °C. The HSC cell line and DU-145 cell line were obtained from Dr. Yeying Wang. The SJSA-1 cell line and HT-29 cell line were provided by lab of Dr. Suzheng Dong.

## Data Availability

Data is contained within the article or Appendix A.

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
