# Peer review of "Design, Synthesis, and Evaluation of Near-Infrared Fluorescent Molecules Based on 4H-1-Benzopyran Core"

_molecules, 2021, doi:10.3390/molecules26226986_

Round 1

Reviewer 1 Report

In this manuscript, the authors reported the design, synthesis, and spectroscopic properties of a series of coumarin-analogues that emit in the red and near-infrared region. Steady-state electronic absorption and emission properties of the chromophores were studied in solvents varying dielectrics as well as viscosities. Cell experiments also suggested that the chromophores have potentials in tumor cell imaging.

Comments:

  • Figures 2-5 need to be expanded since it’s hard to read the details.
  • In the introduction, what’s the relationship between the unique characteristics of electrons and the development of the D-A fluorescence molecules?
  • Please define D-A when it first appears.
  • In the introduction, the authors state that “in recent years, the focus was switched to other cores, including the coumarins”. However, as far as I am concerned, coumarin has a much longer history since its discovery than above mentioned BODIPY dye. They authors are encouraged to elaborate more on this statement.
  • On line 114, what does it mean by “M-3 showed no optical ability in PBS and H2O”? Does it mean it won’t emit or it doesn’t go into water or PBS at all?
  • Please show whole spectra for Figure 3A and 3B that extend into longer wavelength in the main body of the paper or in the SI for reference.
  • On Line 122, please elaborate more on the statement “although the fluorescent test didn’t fully overcome aggregation quenching…”. What’s the degree of aggregation depending on different concentration and solvents? Additional experiments on different concentrations are recommended since aggregation is an issue here.
  • Experimental details for the BSA binding are missing for Figure 4B and Figure 5C. The authors are encouraged to elaborate how the binding conducted and what are the separation methods to get rid of the aggregated dye on the protein. Also, reference is needed for the claiming “According to the previous data…”
  • Detail experimental sections for determine the fluorescence quantum yields are missing.

In summary, this manuscript needs to be reconsidered after major revisions before it can be recommended to publish in Molecules.

Author Response

Dear Editor,

We would like to thank the editor for giving us a chance to revise the paper, and also thank the reviewers for giving us constructive suggestions which would help us both in language and in depth to improve the quality of the paper. Here we re-submit a new version of our manuscript (Manuscript ID: molecules-1455680,Title:"Design, Synthesis, and Evaluation of Near-infrared Fluorescent Molecules Based on 4H-1-Benzopyran Core"), which has been modified according to the editor’s and reviewers’ suggestions.

Sincerely yours,

Mingliang Ma

The following is a point-to-point response to the comments.

Review 1:

In this manuscript, the authors reported the design, synthesis, and spectroscopic properties of a series of coumarin-analogues that emit in the red and near-infrared region. Steady-state electronic absorption and emission properties of the chromophores were studied in solvents varying dielectrics as well as viscosities. Cell experiments also suggested that the chromophores have potentials in tumor cell imaging.

Comments are as follows:

  1. Figures 2-5 need to be expanded since it’s hard to read the details.

Answer:

Thank you for your constructive comments and we have revised them. Figure 2-5 have been replaced by high resolution figures.

  1. In the introduction, what’s the relationship between the unique characteristics of electrons and the development of the D-A fluorescence molecules? Please define D-A when it first appears.

Answer:

Thank you for your constructive comments. The occurrence of fluorescence is closely related to the electronic transition, and the detailed information has been added in Line 35. " When it comes to D-A-D or D-A structure, electrons transition freely in the entire con-jugation interval,and as the effective conjugation area expands, the energy band gap decreases, which would change the fluorescence characteristics of target compounds. Therefore, through the modification of the conjugated skeleton or terminal substituents, different application requirements of fluorescent molecules could be met. [5]"([5]- doi:10.1016/j.dyepig.2020.108596.)

  1. In the introduction, the authors state that “in recent years, the focus was switched to other cores, including the coumarins”. However, as far as I am concerned, coumarin has a much longer history since its discovery than above mentioned BODIPY dye. They authors are encouraged to elaborate more on this statement.

Answer:

Thank you for your constructive comments. According to the literature 【2-5】, there were many fluorescent molecules about the coumarin mother nucleus. we assumed that the deviation is that the coumarin nucleus is relatively new, and the response error has been corrected in Line 43. Thanks again for pointing out the problem in the manuscript.

【2】Torres-Moya I.; Carrillo J.R.; Gomez M.V.; Velders A.H.; Donoso B.; Rodríguez A.M.; Díaz-Ortiz A.; Navarrete J.T.L.; Ortiz R.P.; Prieto P. Synthesis of D-π-A high-emissive 6-arylalkynyl-1,8-naphthalimides for application in Organic Field-Effect Transistors and optical waveguides. Dyes and Pigments. 2021, 191, doi: 10.1016/j.dyepig.2021.109358.

【3】Ding, G.-Y.; Zang C.-X.; Zhang H.; Su Z.-M.; Li G.-F.; Wen L.-L.; Han X.; Xie W.-F.; Shan G.-G. Administration of the D-A structure and steric hindrance effect to construct efficient red emitters for high-performance OLEDs with low efficiency roll-off. Dyes and Pigments. 2021, 192, doi: 10.1016/j.dyepig.2021.109398.

【4】Ipek, O.S.; Topal S.; Topal S.; Ozturk T. Synthesis, characterization and sensing properties of donor-acceptor systems based Dithieno [3,2-b;2′,3′-d] thiophene and boron. Dyes and Pigments. 2021, 192, doi: 10.1016/j.dyepig.2021.109458.

【5】Kunyan Wang; Yongqi Bao; Senqiang Zhu; Rui Liu; Hongjun Zhu. Novel 1,5-naphthyridine-chromophores with D-A-D ar-chitecture: Synthesis, synthesis, luminescence and electrochemical properties. Dyes and Pigments, 2020,181 ,doi:10.1016/j.dyepig.2020.108596.

  1. On line 114, what does it mean by “M-3 showed no optical ability in PBS and H2O”? Does it mean it won’t emit or it doesn’t go into water or PBS at all?

Answer:

Thank you for your constructive suggestions According to the raw data, the M-series compounds were soluble in water and PBS, and the corresponding ultraviolet absorption wavelength and fluorescence emission wavelength were measured. Due to the limited scale of the x-axis of the graph, compared with other organic solvents, the fluorescence emission intensity of M-3 in water and PBS is relatively low, and it is a bulge with a larger slope. PBS has weak optical capability.

  1. Please show whole spectra for Figure 3A and 3B that extend into longer wavelength in the main body of the paper or in the SI for reference.

Answer:

Thank you for your constructive comments. And we have revised them. The wavelength in Figure 3A and 3B have been fully showed in the new Figure 3A and 3B in manuscript. We Set range for fluorescence spectrum test whin 550-805nm.

  1. On Line 122, please elaborate more on the statement “although the fluorescent test didn’t fully overcome aggregation quenching…”. What’s the degree of aggregation depending on different concentration and solvents? Additional experiments on different concentrations are recommended since aggregation is an issue here.

Answer:

Thank you for your constructive comments. Although the compound does not show superior fluorescence emission characteristics in water and PBS, it has a certain fluorescence emission intensity in water and PBS. It is mentioned in 5, but it is much weaker than the emission intensity in organic solvents. It is determined that M-3 has a strong aggregation quenching phenomenon in water and PBS. Before finishing this work, we did do the fluorescence test of M-3 at different concentrations of 100nM, 10mM, 20mM, 50mM, but due to the subtle differences, the result was not added to the manuscript. The compounds didn’t t completely overcome aggregation quenching in water, but that did not affect the entry of M-3 into the cell to bind to proteins.

  1. Experimental details for the BSA binding are missing for Figure 4B and Figure 5C. The authors are encouraged to elaborate how the binding conducted and what are the separation methods to get rid of the aggregated dye on the protein. Also, reference is needed for the claiming “According to the previous data…”

Answer:

Thank you for your constructive comments. "According to the previous data" referred to the experimental data based on viscosity sensitivity test. Because the purpose of the viscosity test was to prove that M-3 can penetrate the cell membrane inside the cell; and the affinity binding with BSA is to prove that the compound will bind to the protein after entering the cell. The information has been added to the Line163-165. And the BSA+M-3 binding method was add in Line 166-170.

  1. Detail experimental sections for determine the fluorescence quantum yields are missing.

Answer:

Thank you for your constructive comments. The quantum yield test in this experiment obtains QEin data through integrating sphere device, which was explained in Line 242-250

Reviewer 2 Report

Authors explained synthesis and studied their viscosity and biological studies. the overall manuscript is well explained and the M3 used for tumor imaging. before accepting the manuscript please include these points.  

  1. In figure 1, authors said Design of near-infrared small molecule M-1. but in the figure its mentioned M3. authors need to change accordingly.
  2. the reference 26 is not matching with the content. need to be recheck all the references. 
  3. figure 2 is blurry and not visible properly. so need to change with high resolution figure.
  4. in figure 2, there is no naming of the materials like M1, M2 and M3. what is WST-2-39, WST-2-41 and WST-2-45. authors need to check.
  5. in line 91, it should be ΔEST.
  6. authors need to mention abbreviations of QEin and AF (Table 1) in manuscript.
  7. figure 6 also need to replace with high resolution image.
  8. authors need to include theoretical studies like which method, basis set and software.. etc., in experimental session.
  9. authors need to include optimized geometry of all materials and their xyz coordinates in SI.
  10. authors need to give references for DFT and TD-DFT methods and for software tey used.
  11. why authors only Viscosity sensitivity studies of M-3, aht about M1 and M2. if studied Viscosity studies need to mention in SI.

Author Response

Response to reviewers

Dear Editor,

We would like to thank the editor for giving us a chance to revise the paper, and also thank the reviewers for giving us constructive suggestions which would help us both in language and in depth to improve the quality of the paper. Here we re-submit a new version of our manuscript (Manuscript ID: molecules-1455680,Title:"Design, Synthesis, and Evaluation of Near-infrared Fluorescent Molecules Based on 4H-1-Benzopyran Core"), which has been modified according to the editor’s and reviewers’ suggestions.

Sincerely yours,

Mingliang Ma

The following is a point-to-point response to the comments.

-Reviewer 2

Authors explained synthesis and studied their viscosity and biological studies. the overall manuscript is well explained and the M3 used for tumor imaging. before accepting the manuscript please include these points. 

Major points

  1. In figure 1, authors said Design of near-infrared small molecule M-1. but in the figure its mentioned M3. authors need to change accordingly.

Answer:

Thank you for your constructive comments and we have revised The annotation of Figure 1 has been replaced as M-3.

  1. the reference 26 is not matching with the content. need to be recheck all the references.

Answer:

Thank you for your constructive suggestions. The reference 26 has be corrected as [27]- Wenjian Liu; Yunlong Xiao. Relativistic time-dependent density functional theories. Chemical Society Reviews, 2018, 47, 4481, doi:10.1039/c8cs00175h.]

  1. figure 2 is blurry and not visible properly. so need to change with high resolution figure.

Answer:

Thank you for your constructive suggestions. Figure 2-6 have been replaced by high resolution figures.

  1. in figure 2, there is no naming of the materials like M1, M2 and M3. what is WST-2-39, WST-2-41 and WST-2-45. authors need to check.

Answer:

Thank you for your constructive suggestions. We have revised the compounds names in Figure 2.

  1. in line 91, it should be ΔEST.

Answer:

Thank you for your constructive suggestions and we have revised as ΔEST.

  1. authors need to mention abbreviations of QEin and AF (Table 1) in manuscript.

Answer:

Thank you for your constructive suggestions. The quantum yield test in this experiment obtains QEin a through integrating sphere device, which was explained in Line 279-287.

  1. figure 6 also need to replace with high resolution image.

Answer:

Thank you for your constructive suggestions. Figure 2-6 have been replaced by high resolution figures.

  1. authors need to include theoretical studies like which method, basis set and software.. etc., in experimental session.

Answer:

Thank you for your constructive suggestions. Method, basis set and software of orbital calculation were added in Materials and Methods session.

  1. authors need to include optimized geometry of all materials and their xyz coordinates in SI.

Answer:

Thank you for your constructive suggestions. Optimized geometry of M-1, M-2, ere zipped as” M-1-3_ES_GS_MOL-SI.zip” which would be uploaded independently. And xyz coordinates of M-1, M-2, and M-3 was added as Table S8 in SI.

  1. authors need to give references for DFT and TD-DFT methods and for software tey used.

Answer:

  1. Thank you for your constructive suggestions. Frontier orbital calculations were performed using Gaussian 16 software. Detailed information was mentioned in Line 259-264, and reference was added to [39](M. J. Frisch, G. W. Trucks, H. B. Schlegel, G. E. Scuseria, M. A. Robb, J. R. Cheeseman, G. Scalmani, V. Barone, G. A. Petersson, H. Nakatsuji, X. Li, M. Caricato, A. V. Marenich, J. Bloino, B. G. Janesko, R. Gomperts, B. Mennucci, H. P. Hratchian, J. V. Ortiz, A. F. Izmaylov, J. L. Sonnenberg, D. Williams-Young, F. Ding, F. Lipparini, F. Egidi, J. Goings, B. Peng, A. Petrone, T. Henderson, D. Ranasinghe, V. G. Zakrzewski, J. Gao, N. Rega, G. Zheng, W. Liang, M. Hada, M. Ehara, K. Toyota, R. Fukuda, J. Hasegawa, M. Ishida, T. Nakajima, Y. Honda, O. Kitao, H. Nakai, T. Vreven, K. Throssell, J. A. Montgomery, Jr., J. E. Peralta, F. Ogliaro, M. J. Bearpark, J. J. Heyd, E. N. Brothers, K. N. Kudin, V. N. Staroverov, T. A. Keith, R. Kobayashi, J. Normand, K. Raghavachari, A. P. Rendell, J. C. Burant, S. S. Iyengar, J. Tomasi, M. Cossi, J. M. Millam, M. Klene, C. Adamo, R. Cammi, J. W. Ochterski, R. L. Martin, K. Morokuma, O. Farkas, J. B. Foresman, and D. J. Fox, Gaussian, Inc., allingford Gaussian 16, Revision A.03, 2016.)

  1. why authors only Viscosity sensitivity studies of M-3, aht about M1 and M2. if studied Viscosity studies need to mention in SI.

Answer:

Thank you for your constructive suggestions. "Spectral properties of compound M series", line 109,“”ntioned the reason why M-3 was chosen as the compound in further in vitro experiments. So only M-3 was observed in biological experiments. Viscosity sensitivity studies was used to show the ability of the compound to cross the cell membrane into the cell interior, paving the way for imaging. So, there was no viscosity sensitivity studies of M-1 and M-2.

Round 2

Reviewer 1 Report

The revised version is recommended for publication in Molecules as a full article.